# Machine Learning Based on Morphological Features Enables Classification of Primary Intestinal T-Cell Lymphomas

**DOI:** 10.3390/cancers13215463

**Published:** 2021-10-30

**Authors:** Wei-Hsiang Yu, Chih-Hao Li, Ren-Ching Wang, Chao-Yuan Yeh, Shih-Sung Chuang

**Affiliations:** 1aetherAI, Co., Ltd., Taipei 115, Taiwan; seanyu@aetherai.com; 2Department of Pathology, Chang Gung Memorial Hospital, Kaohsiung 83301, Taiwan; s12464@cgmh.org.tw; 3Department of Pathology and Laboratory Medicine, Taichung Veterans General Hospital, Taichung 40705, Taiwan; rcwang@vghtc.gov.tw; 4Department of Nursing, College of Nursing, HungKuang University, Taichung 1018, Taiwan; 5Department of Pathology, Chi-Mei Medical Center, Tainan 71004, Taiwan

**Keywords:** artificial intelligence, digital pathology, quantitative morphology, primary intestinal T-cell lymphoma, convolutional neural network, human-interpretable AI, monomorphic epitheliotropic intestinal T-cell lymphoma, peripheral T-cell lymphoma, not otherwise specified

## Abstract

**Simple Summary:**

We presented a machine learning approach for accurate quantification of nuclear morphometrics and differential diagnosis of primary intestinal T-cell lymphomas. The human interpretable machine learning approach can be easily applied to other lymphomas and potentially even broader disease categories. This approach not only brings deeper insights into lymphoma phenotypes, but also paves the way for future discoveries concerning their relationship with disease classification and outcome.

**Abstract:**

The aim of this study was to investigate the feasibility of using machine learning techniques based on morphological features in classifying two subtypes of primary intestinal T-cell lymphomas (PITLs) defined according to the WHO criteria: monomorphic epitheliotropic intestinal T-cell lymphoma (MEITL) versus intestinal T-cell lymphoma, not otherwise specified (ITCL-NOS), which is considered a major challenge for pathological diagnosis. A total of 40 histopathological whole-slide images (WSIs) from 40 surgically resected PITL cases were used as the dataset for model training and testing. A deep neural network was trained to detect and segment the nuclei of lymphocytes. Quantitative nuclear morphometrics were further computed from these predicted contours. A decision-tree-based machine learning algorithm, XGBoost, was then trained to classify PITL cases into two disease subtypes using these nuclear morphometric features. The deep neural network achieved an average precision of 0.881 in the cell segmentation work. In terms of classifying MEITL versus ITCL-NOS, the XGBoost model achieved an area under receiver operating characteristic curve (AUC) of 0.966. Our research demonstrated an accurate, human-interpretable approach to using machine learning algorithms for reducing the high dimensionality of image features and classifying T cell lymphomas that present challenges in morphologic diagnosis. The quantitative nuclear morphometric features may lead to further discoveries concerning the relationship between cellular phenotype and disease status.

## 1. Introduction

Primary intestinal T-cell lymphomas (PITLs) are rare and therefore pose a challenge for diagnosis. According to the 2017 World Health Organization (WHO) classification of lymphoid neoplasms, PITLs include enteropathy-associated T-cell lymphoma (EATL), monomorphic epitheliotropic intestinal T-cell lymphoma (MEITL), intestinal T-cell lymphoma, not otherwise specified (ITCL-NOS), and indolent T-cell lymphoproliferative disorder of the gastrointestinal tract [1]. EATL, previously designated type I EATL, is celiac disease-associated, with a specific genetic predisposition and pathological features (enteropathy, large tumor cells frequently expressing CD30), which is not a diagnostic problem in specific clinical settings [1]. MEITL, previously called type II EATL, is characterized by monomorphic small- to medium-sized tumor cells with an epitheliotropic growth pattern and the expression of CD8, CD56, and cytotoxic markers [2,3,4]. ITCL-NOS is a diagnosis of exclusion and comprises a heterogeneous group of PTCLs that do not meet the diagnostic criteria for any of the specific types of PITL [1,5]. The neoplastic cells of ITCL-NOS are usually medium to large-sized and pleomorphic, with a cytotoxic phenotype [6]. Due to the inconsistency between phenotypic and morphological features of PITLs, it might not be easy to make a straightforward distinction between MEITL and ITCL-NOS in some cases. For example, there are cases showing a METIL immunophenotype but with a more pleomorphic nuclear feature or cases showing a MEITL cellular morphology but with an atypical immunophenotype. These cases with an inconsistency between the immunophenotype and the cellular morphology patterns were regarded as borderline in the study. In practice, a major challenge for pathological diagnosis is the distinction between MEITL and ITCL-NOS, as the major diagnostic criterion relies on morphological evaluation.

The traditional diagnostic approach relying on morphological evaluation may be subjective, with relatively low consistency among pathologists, particularly reflected in the evaluation of megakaryocytic atypia in myelodysplastic/myeloproliferative neoplasms. Recently, emerging technologies such as deep neural networks, especially the convolutional neural network (CNN) technique, demonstrate superior performance for classification, segmentation, and detection tasks in digital pathology image analysis [7]. For example, Banaeeyan et al. [8] proposed a deep-learning-based semantic segmentation algorithm to identify tumor budding on H&E-stained whole-slide images (WSIs) of colorectal cancer. Similarly, Nateghi et al. [9] proposed a deep-learning-based approach for the detection of mitoses on WSIs of breast cancers and achieved a superior performance than all other previous approaches. Despite its success in various pathology image recognition tasks, we reasoned that an end-to-end approach, i.e., training CNN to directly classify lymphomas, offers only very limited help to physicians. Instead, using CNN’s superior image recognition ability to generate human interpretable features, such as nuclear morphometrics, may not only reduce the high dimensionality of image features [10,11,12] but also deepen physicians’ understanding of the diseases, which would then lead to more precise diagnosis. Nuclear segmentation is critical in digital pathology and enables the extraction of cellular morphometric features for quantitative analysis. Once nuclei are segmented precisely, cell morphometric features such as size, smoothness, intensity entropy, and pleomorphism can be easily derived, not only to assess the disease grades but also to quantify morphological indexes into spectrums for disease differentiation and severity classification [13,14,15,16].

The aim of this study was to investigate the feasibility of using machine learning (ML) techniques based on quantitative cell morphometric information to classify histopathology images between the two subtypes of PITLs, namely, MEITL versus ITCL-NOS. Firstly, a deep-learning-based algorithm was applied to segment lymphocytes, followed by the computation of cell morphometrics. Then, XGBoost, a decision-tree-based machine learning algorithm, was trained to classify PITL cases based on the computed nuclear morphometrics [17]. Our results demonstrate that: (1) deep learning algorithms could accurately segment nuclear boundaries, and the morphometric features could be quantified; (2) there were significant differences in quantitative morphological features between the neoplastic cells of MEITL and ITCL-NOS; (3) The XGBoost model based on the extracted morphological profile showed a superior performance to CNN applied directly to images in classifying MEITL from ITCL-NOS, and achieved an area under receiver operating characteristic curve (AUC) of 0.966.

## 2. Materials and Methods

### 2.1. Patient Samples

A total of 40 PITL patients, including MEITL (*n* = 26), ITCL-NOS (*n* = 10), and borderline cases (*n* = 4), were collected from 17 hospitals in Taiwan. All 40 specimens were surgical specimens and their diagnoses were made by a senior hematopathologist (SSC) based on the 2017 WHO criteria [1]. They were all PITL cases, as confirmed by staging workups. All cases were negative for Epstein–Barr virus (EBV) as assessed by in situ hybridization for EBV-encoded small RNA (EBER). Some cases had been included in our previous study on the significance of EBV in PITL [6]. All four cases (no. T02, T13, T18 and T20) examined for serum anti-tissue transglutaminase by ELISA were negative [6]. The major clinicopathological findings of the four borderline cases are summarized in Table 1. Phenotypically, they were all consistent with MEITL with the expression of CD8, CD56, and cytotoxic markers; however, the tumor cells were larger than that would be expected in typical MEITL cases. The study was approved by the Institutional Review Board at Chi-Mei Medical Center (approval no. 10612-010).

### 2.2. Dataset Preparation

For each patient, one hematoxylin-and-eosin-stained section was scanned and digitalized to create a total of 40 WSIs with an average size of 175,274 × 407,126 pixels. To avoid image areas of inferior quality (due to conditions such as out-of-focus scanning, tissue folding, and bubbles), the slides were reviewed by a senior hematopathologist (S.S.C.), who manually selected the most representative and high-quality area in each slide before scanning (a representative circled area as shown in Figure 1A). The sections were scanned using the Pannoramic 250 Flash digital slide scanner (3DHISTECH, Hungary) using 40× objective. For each marked area, 33 regions of interest (ROI, size = 115 μm × 115 μm) randomly sampled from 19 cases were extracted for the development of the lymphocyte detection model, whereas 10 high-power fields (HPF, size = 346 μm × 346 μm) from each case were randomly selected (green boxes in Figure 1A) for the development of nuclear morphometric feature exaction and the lymphoma classification model.

For the lymphocyte detection model, a full dataset of 33 ROIs were split into the training, validation, and testing sets at a ratio of 8:1:1. ROIs used as testing sets were contained in WSIs independent from those that contained training and validation ROIs. Within each ROI, the contours of lymphocytes were manually annotated by two pathologists (C.H.L. and S.S.C.). For the development of the disease classification model, a total of 400 HPFs were used as the dataset. A 3-fold cross-validation was employed to train and evaluate the model such that the images from an individual patient would only appear in either the training or validation set. The four borderline cases were excluded from both the training and testing sets and were used only as references. The classification model was tested at the case level. The recruited cases and the disposition of training and validation sets under different phases are summarized in Table 2.

### 2.3. Lymphocyte Detection Model

To segment the neoplastic lymphocytes, an instance segmentation model was trained by employing a region-based CNN model—the hybrid task cascade regional proposal convolutional network (HTC-RCNN) [18] with ResNet50 [19] as the backbone. The HTC-RCNN model was trained using the stochastic gradient descent (SGD) optimizer, at a learning rate of 0.001, and a batch size of 16 on a single NVIDIA V100 GPU. The performance of the detection model was evaluated using average precision (AP). To compute AP, each detected box was first matched to the ground truth to evaluate whether the intersection-over-union (IoU) was over a threshold of 0.5. Subsequently, precision scores were estimated under different recall thresholds ranging from 0 to 1, and AP was computed by summing the area under the precision-and-recall curve.

### 2.4. Computation of Nuclear Morphometrics

The trained lymphocyte detection model was applied to the 400 HPFs to segment lymphocytes. For quantitative analysis of nuclear morphology, seven numerical attributes were calculated for each segmented nucleus of lymphocyte (Table 3). For each HPF, four moment statistics including mean, variance, skewness, and kurtosis were computed for the whole cellular population within an HPF. Thus, the nuclear morphology of each HPF was characterized by 28 features. The throughput of our morphology extraction module was estimated, and it required an average of 2.22 ± 0.42 s (range: 0.91 to 3.13 s) to process each HPF, which could be applicable for clinical practice. At the case level, the extracted features were further aggregated by averaging feature scores across HPFs for each case no. to form a feature set (feature matrix). The feature set was used in a decision-tree-based machine learning algorithm (XGBoost) for disease type classification at the case level.

### 2.5. T-cell Lymphoma Classification Model

We used extreme gradient boosting (XGBoost), a tree-based machine learning approach, to select important features for modeling case-level classification. For the machine learning algorithm, the weight by Gini was applied as the feature selection method iteratively to alleviate the effect of redundant features and to minimize the loss of binary cross-entropy, while random search was employed as the hyper-parameter optimization approach. Two XGBoost models were employed, taking different input sets: (1) only morphological features (28 feature scores per subject) were used as inputs; (2) both morphological features and immunohistochemical (IHC) phenotypes (i.e., either positive or negative for the expression of CD8 and CD56) were used as inputs. A binary classifier was trained to classify each HPF into two types of PITLs, either MEITL or ITCL-NOS, based on the selected features.

To compare the model performance of the XGBoost approach versus the CNN approach, a CNN with ResNet50 backbone was also trained by applying the AdamW optimizer, at a learning rate of 0.001, with a batch size of 16 using an Nvidia Quadro RTX 8000. To prevent the loss of the model in local minima due to limited number of training images, the AdamW optimizer is preferable to the SGD in the current study. The data augmentation technique was applied during the training phase to increase data variability. The applied augmentation methods in this study included random horizontal/vertical flipping, random translation, random rotation, random scaling, random color jittering in brightness/contrast/saturation/hue, random Gaussian blurring, and random cropping. An early-stopping technique was employed to stop training processing when validation loss did not improve for 10 epochs. The CNN took an image of HPF as the input and predicted the disease type of the image based on its case-level outcome. For the testing phase, the case-level prediction for disease type was based on the classification decisions (outcome probabilities) averaged from the 10 HPFs of each case. The classification performance of different models was evaluated using the area under the receiver-operating curve (AUC) with a 3-fold cross-validation, where each fold is split at the case level. Finally, Delong’s test was employed to compare AUCs between models with a two-tailed hypothesis testing at the significance level of 0.05.

In addition to the case-level prediction, the feature importance of the XGBoost was evaluated using “gain”, which implies how informative the features are when used to split the data across all trees. Higher “gain” indicates a greater contribution of the corresponding feature to the model. 

### 2.6. Modeling for Diagnostic Prediction from the Feature Profile

A general linear model (GLM) was employed to better understand the effects of different morphological features on diagnostic decisions. A repeated measurement method was used to control the within-subject variations. Given the fact that the cells sampled from a single HPF might not be fully representative of each case, 10 HPFs were randomly sampled and used to estimate the error variance sourced from HPF sampling. The effects of the morphological feature scores on the prediction of the diagnosis can be formulated as follows: $Y_{ij}=\mu +\alpha_{j}+\pi_{i}+\epsilon_{ij}$, where *Y_ij_* (dependent variable) denotes the morphological feature score; *μ*, the population mean of the feature; *α_j_*, the group mean score sourced from disease type *j*; *π_i_*, the mean score of *Case_i_*; and $\epsilon_{ij}$, the residual error for the observation. The four borderline cases were not included in the statistical analysis.

The descriptive statistics of the 28 aggregate morphological feature measurements of two lymphoma types (METIL vs. ITCL-NOS) and the F-test were conducted to test the differences of each feature score between two disease subtypes.

## 3. Results

### 3.1. The Lymphoma Nucleus Detection Model Shows High Sensitivity along with High Positive Predictive Value

An average of 892.18 ± 254.80 nuclei (range: 331 to 1502) were detected in each HPF. The lymphoma nucleus detection model achieved an average precision of 0.943 and 0.881 on segmenting lymphoma nuclei for the validation set and the testing set, respectively. For the testing set, our model achieved a precision (positive predictive value) of 0.911 with a recall (sensitivity) of 0.868. Representative examples of boundary-segmented lymphoma nuclei with monomorphic, pleomorphic, and borderline morphology were depicted in Figure 1B.

### 3.2. The T-Cell Lymphoma Classification Model Discriminated MEITL and ITCL-NOS Cases and Showed Higher Accuracy Than the CNN

The classification performance of two XGBoost models using morphological feature measurements with or without IHC phenotypes as inputs on discriminating the two types of PITLs was evaluated by using AUC. As depicted in Figure 2A, the XGBoost model using only morphological feature measurements achieved an AUC of 0.966 (95% CI: 0.949–0.984), while the XGBoost model using both morphological feature measurements and IHC phenotypes as inputs achieved an AUC of 0.955 (95% CI: 0.935–0.975), indicating no significant difference in the discriminative power by adding the IHC phenotype as a model input (*p*-value = 0.412), as assessed by Delong’s test. In comparison, the deep CNN directly trained using HPFs as image inputs achieved an AUC of 0.820 (0.734–0.906), which was significantly inferior to the XGBoost method (without IHC, *p* < 0.01; with IHC, *p* < 0.01; Table 4).

As shown in Figure 3, there was a high concordance between the XGBoost model and the diagnoses by the senior hematopathologist except for case nos. T02, T20, and T70. Case no. T20 was diagnosed as ITCL-NOS by the hematopathologist but was predicted as MEITL by the XGBoost model; while both cases no. T02 and T70 were diagnosed as MEITL but predicted to more likely be ITCL-NOS by the model.

### 3.3. The Importance of Features Obtained from the XGBoost Model Can Be Ranked

As illustrated in Figure 2B, the most important morphological features selected based on the “gain” derived from the XGBoost model included the variance in perimeter, the variance in nuclear area, and the mean of nuclear irregularity. When the IHC phenotype was used as an input in the XGBoost model, both CD56 and CD8 expression status ranked higher than most of the other morphological features, except for the variance in perimeter, indicating that the variance in perimeter contributed the most to the classification decisions.

### 3.4. Feature Analysis Using the GLM Enabled Explicit Interpretation of the Morphological Features

Cell contour delineated by the detection model enabled the quantitative comparison of the morphological profiles between the two disease subtypes. The 28 morphological feature scores computed for each case are presented in Table 5. These features were used to conduct statistical analysis using GLM. Among all the detected cells, the average cell size was 41.72 ± 9.00 μm^2^; the axis ratio, 1.29 ± 0.06; the perimeter, 24.32 ± 2.66 μm; the irregularity, 1.15 ± 0.40; the circularity, 0.75 ± 0.03; and the entropy, 5.85 ± 0.10, respectively.

The *F*-statistic derived from GLM was employed to access the significance of variance between the means of the two disease subtypes among morphological features measurements. Generally, the *F*-test estimates if the two effects are sourced from the same population. A two-tailed null hypothesis was employed to examine the significance. The results reveal statistically significant differences in several morphological feature measurements between the two disease subtypes. For differentiating MEITL and ITCL-NOS, five out of seven measurements in variance were significant (*p* < 0.01) and five of seven measurements in mean were significant. Only irregularity and perimeter showed a significant difference in both measurements in skewness and kurtosis. The orientation of cells in the local region showed no statistical significance in differentiating these two diseases. Features associated with the size of cells, including the perimeter and the area, showed a stronger effect (*p* < 0.001) for measurement in the variance than in the mean.

As illustrated in Figure 4B, the two different disease subtypes can be visually separated by plotting the variance in nuclear perimeter versus the variance in nuclear irregularity. The borderline cases (green dots) that could be hardly distinguishable were also shown in Figure 3B. For MEITL cases, the variance in perimeter and the variance in irregularity were smaller and showed a linear correlation; on the contrary, the ITCL-NOS cases showed a higher variance in irregularity or perimeter. 

### 3.5. The Model Produced a 1:1 Ratio Prediction for the Four Borderline Cases

The clinicopathological findings and the model prediction results of the four borderline cases as diagnosed by the senior hematopathologist are presented in Figure 3B. Phenotypically, they were all consistent with MEITL, with neoplastic cells expressing CD8, CD56, and cytotoxic markers; however, the tumor cells were larger than would be expected in typical cases of MEITL. Based on the model prediction, two cases (case nos. T31 and T63) were predicted as MEITL; while the other two cases (T05A and T60) were predicted as ITCL-NOS.

## 4. Discussion

Due to the rarity of PITL, differentiating MEITL from ITCL-NOS is challenging for pathologists, and the distinction relies on the experience of the pathologists in evaluating histopathological features, particularly monomorphic versus pleomorphic tumor cells, in conjunction with immunophenotype; nevertheless, the morphologic classification by pathologists could be subjective. In the current study, we collected PITL cases from 17 hospitals in Taiwan and demonstrated a successful workflow that could detect and segment lymphoma nuclei accurately using a CNN model, followed by knowledge-based morphological feature extraction for quantitative analysis and classification of PITL using an XGBoost model. 

Previous studies have shown that analyzing cellular properties such as nuclear size and the homogeneity of the cell population can be clinically useful for differentiating different groups of cells and for predicting prognosis. For example, Maqlin et al. [15] and Faridi et al. [14] applied boundary-based and region-growing methods to segment cells for extracting morphological features such as nuclear size, solidity, and eccentricity to grade pleomorphism scores of WSIs of breast cancer cases, and achieved an accuracy of 89.4% and 86.6%, respectively. Similarly, Moran et al. [16] employed the cell detection module in QuPath [20], an open-source image analysis toolbox for digital pathology, to extract the morphological features of Markel cell carcinoma (MCC) and found that nuclear area and circularity were crucial factors for the prognosis of MCC. To date, many critical findings have been achieved by analyzing morphological features using traditional image-processing methods with handy packages such as QuPath and ImageJ; however, the selection of parameters such as background radius, nucleus radius, or local intensity threshold require careful adjustment when processing each single image. As most image-processing methods adopt the same hyper-parameter set to segment cells, it may fail in some challenging cases, including: (1) image with crowded cells, (2) cells with inconsistent shape or size in a local area, and (3) cells with hyperchromatic or vesicular nucleus [21].

The state-of-the-art deep-learning-based segmentation models have achieved excellent performance in nuclear segmentation [22,23]. As illustrated in Figure 5, QuPath with the default setting can segment the cell boundary successfully in most cases; however, fragile contours and false positives occurred frequently when cells were crowded, vesicular, or with atypical chromatin patterns. In comparison, our deep-neural-network-based algorithm showed a robust segmentation result with less fragile contours. Notably, nuclei of other types of cells, such as vascular endothelial cells, may fail to be distinguished from lymphocytes because only lymphocytes were labelled to train the detection model in the current study. Falsely included non-lymphoid cells may slightly distort the estimation of morphological features. The inclusion of various types of annotated cells are expected to improve the algorithm. Still, using CNNs to segment cells precisely, requiring no tedious parameter adjustment across different fields of view, could enable pathologists to analyze nuclear morphology without manual region selection and can be applied to WSIs efficiently.

In accordance with the aforementioned studies using nuclear morphometrics for classifying cells into benign versus malignant [24,25,26] or presence versus absence of Merkel cell polyoma virus in MCC [27], the current study demonstrated that nuclear morphological patterns made a significant contribution in differentiating MEITL from ITCL-NOS. Specifically, the variance in perimeter and irregularity were two of the most critical features to separate MEITL and PICL-NOS into different clusters. As illustrated in Figure 4B, the shape of lymphocytes in a typical MEITL case (case no. T48) was apparently round and regular, while that in a typical ITCL-NOS case (case no. T45A) was pleomorphic. For atypical cases, such as case nos. T02 and T69 (Figure 4B), it was difficult to classify them into either MEITL or ITCL-NOS according to the appearance of nuclei (in the middle of both slides) by human eyes. Using our algorithm-derived morphological indexes and classification scheme, the four borderline cases as judged by the experienced hematologists (typical immunophenotype but with more pleomorphic nuclei than typical MEIL cases) were classified either as MEITL (case nos. T31 and T63) or ITCL-NOS (nos. T05 and T60), indicating limitations from both manual and AI-assisted histopathological evaluation when it comes to making a clear-cut diagnosis. Alternatively, the degree of “monomorphism” might not be as strict as the name of MEITL implies, suggesting that there might be a certain degree of nuclear pleomorphism in cases currently defined as MEITL. The nature of such borderline cases in our series may potentially be revealed by their underlying genetic features, which we have been currently investigating.

Immunophenotypically, Weisenburger et al. [28] reported that 19% and 6% of ITCL-NOS cases expressed CD8 and CD56, respectively, indicating a phenotypic overlap between MEITL and ITCL-NOS. Our algorithm could classify PITL cases into either MEITL or ITCL-NOS based on the morphological indexes; however, the relationship between these indexes and disease prognosis remains unclear. Further studies are needed to explore the relationship between morphological features and patient outcomes with these rare tumors. For differentiating MEITL and ITCL-NOS, immunophenotypic features were found to bring no additional benefits for classification performance to the tree-based models on top of morphological features in our models.

Using CNNs as an end-to-end solution to estimate clinical quantification indexes, such as nuclear atypia score [7,13], has become popular in recent years. In this study, the CNN model trained by images directly was capable of differentiating these two types of PITLs (AUC = 0.82); however, the classification performance was significantly inferior to the tree-based models (*p* < 0.01) using quantitative nuclear morphometric features as input. Previous studies have shown that CNNs can achieve over 95% AUC in the classification of lymphomas, for instance, the diffuse large B-cell lymphoma [29] and the follicular lymphoma (FL) [30]. In these studies, the researchers utilized architectural patterns rather than cellular details for analysis. For instance, in our previous study, we found that the duodenal FL had a higher density of follicles and larger follicle size as compared with the reactive lymphoid hyperplasia [31]. In the current study, the growth patterns of PITL, either MEITL or ITCL-NOS, was diffuse and we could not use growth pattern for differential diagnosis. Instead, we needed to extract morphological features of the single tumor nucleus for second-order statistical analysis. Notably, CNNs are known to achieve an inferior performance when there are insufficient training data, whereas XGBoost models with expert-defined features could achieve a satisfactory performance with limited data, because this prevents models overfitted to the ultra-high dimensionality of image features. Incorporating more cases might improve the classification performance of the CNN model. The explainability of CNNs remains a debatable but valuable research topic in the future, especially in medical image analysis. In contrast, models based on accurately segmented-then-extracted handcrafted features could be used as clinical indexes for prediction, given that the weights of each extracted feature parameter can be estimated and understood by pathologists. The weighting of extracted features makes it easier for pathologists to associate their diagnostic experiences with digital findings, and thus, the algorithm may offer the possibility of a reliable computer-aided system for routine diagnostic practice.

While most quantitative indexes, such as mitoses, tumor budding, and the Ki67 index, require pathologists to follow a complex visual scoring protocol to execute laborious counting, using ML algorithms to accomplish an automated scoring is ideal given that a computer algorithm can provide a consistent and objective result efficiently. Previous studies have demonstrated that an ML algorithm is able to achieve comparable quantification results with manual examination by pathologists in a variety of tasks, including cell counting and IHC scoring [32]. In this study, cellular morphology was quantified, and the results show that several extracted morphological indexes, including nuclear size, nuclear circularity, and irregularity, were found to be important morphological features and showed a significant contribution in regression models for differentiating MEITL from ITCL-NOS. These findings are in line with previous findings showing correlations between morphological indexes and the severity of diseases. With the aid of AI, such morphological evaluation and counting tasks can be easily accomplished on a larger scale, which is beyond most of the current visual scoring protocols using limited numbers of cells for the score index, after selecting the representative fields.

In the context of clinical practice, pathologists may incorporate the AI-derived results with immunophenotypic findings for diagnosis, particularly for morphologically borderline cases as shown in our current study. In computational pathology, the ML algorithm may be applied to other indications; for example, to differentiate classic versus pleomorphic cytomorphological variants among mantle-cell lymphomas or to classify low- versus high-grade follicular lymphoma. Therefore, our workflow might be applicable to various tumor types for morphological evaluations.

## 5. Conclusions

In conclusion, this study used ML techniques based on quantitative cell morphometric information for classifying histopathological images of PITLs into two subtypes, either MEITL or ITCL-NOS. The classification performance of the XGBoost model was superior to the end-to-end CNN model and could elucidate explicit relationships between predictions and morphological features. Furthermore, it achieved a comparable result to that of the incorporation of immunophenotype and to that of the senior hematopathologist. Our model may hold great potential to improve diagnostic consistency, efficiency, and accuracy for other types of cancers.

## Figures and Tables

**Figure 1 cancers-13-05463-f001:**
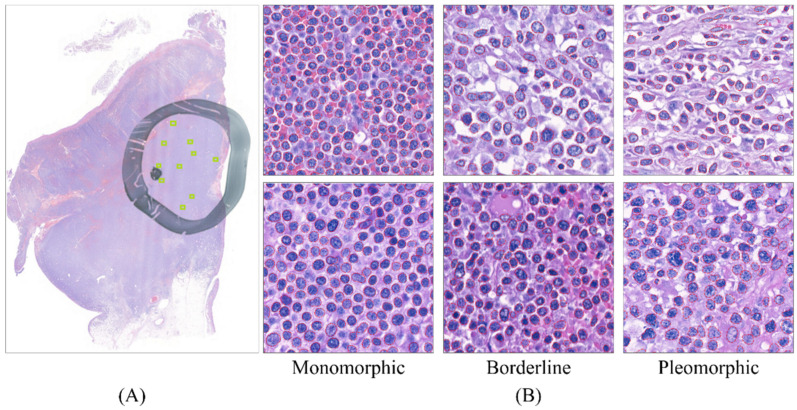
Sample images used as the dataset. (**A**) An example of the whole-slide image. A high-quality region was manually marked by a senior hematopathologist using a black circle. The tiny green boxes represent the randomly sampled high-power fields (HPF). (**B**) Representative HPF images of the neoplastic lymphocytes from MEITL (left), borderline (middle), and ITCL-NOS (right) cases. The neoplastic lymphocytes detected and segmented by the model are contoured in red line.

**Figure 2 cancers-13-05463-f002:**
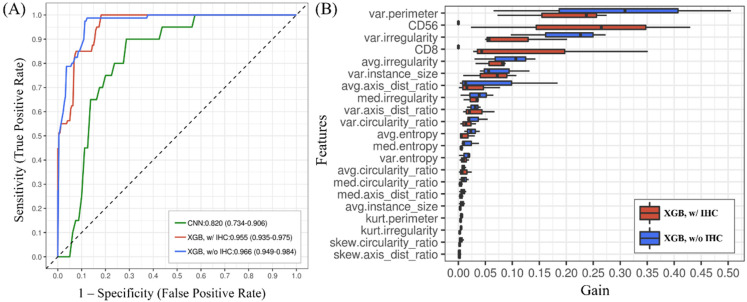
Accuracy and feature analysis. (**A**) Classification accuracy (AUC) of different models. The XGBoost model using only morphological feature measurements (XGB, w/o IHC) achieved an AUC of 0.966 (95% CI: 0.949–0.984), while the XGBoost model using both morphological feature measurements and IHC phenotypes as inputs (XGB, w/IHC) achieved an AUC of 0.955 (95% CI: 0.935–0.975). (**B**) The feature importance of XGBoost models. Higher “gain” indicates a greater contribution of the corresponding feature made to the model. Note: var.=variance; avg.: average; kurt.: kurtosis; skew: skewness; axis_dist_ratio: ratio of axis length; circularity_ratio: circularity; instance_size: nuclear area.

**Figure 3 cancers-13-05463-f003:**
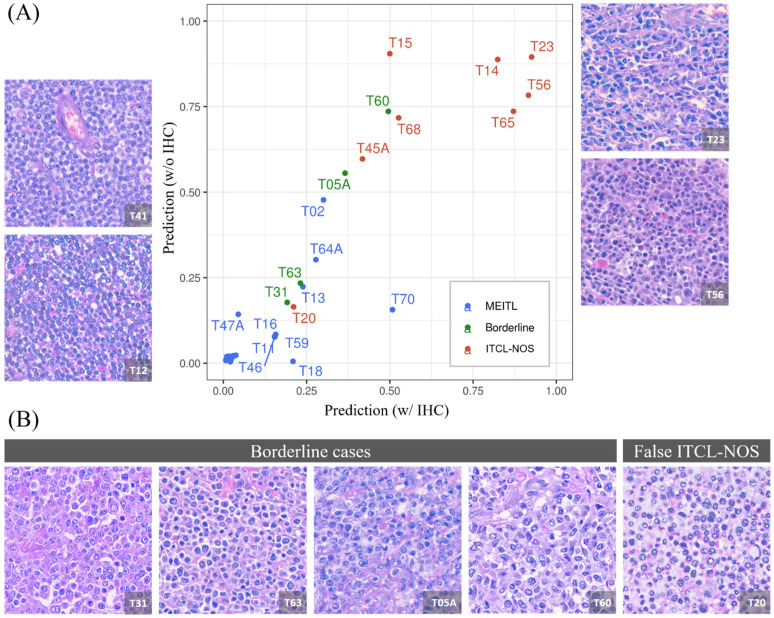
Association between predictions from two XGBoost models. (**A**) The scatter plot of predictions made by the two XGBoost models using morphological features with IHC (w/IHC) versus morphological features without IHC phenotype (w/o IHC). MEITL, ITCL-NOS, and borderline cases are represented by blue, red, and green, respectively. The representative images of typical MEITL (case no. T41 and T12) and ITCL-NOS (case no. T23 and T56) are presented. (**B**) The four borderline cases (case no. T31, T63, T05A, and T60) and a wrongly predicted case (case no. T20, predicted as MEITL but actually ITCL-NOS) are shown in the lower panels.

**Figure 4 cancers-13-05463-f004:**
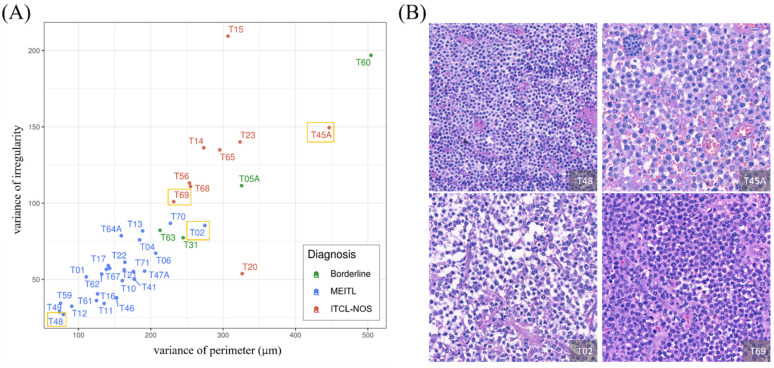
Association between two computed cellular morphometrics. (**A**) The scatter plot of the variance in nuclear perimeter (x-axis) versus the variance in nuclear irregularity (y-axis) of all cases. The MEITL cases, the ITCL-NOS cases, and the borderline cases are represented in blue, green, and red, respectively. (**B**) Representative histological images for MEITL (e.g., case no. T48) are in the lower-left corner of the scatter plot; representative ITCL-NOS cases (e.g., case no. T45A) are shown in the upper-right corner of the scatter plot; atypical cases (e.g., case no. T02 and T69) are located in the middle of the scatter plot.

**Figure 5 cancers-13-05463-f005:**
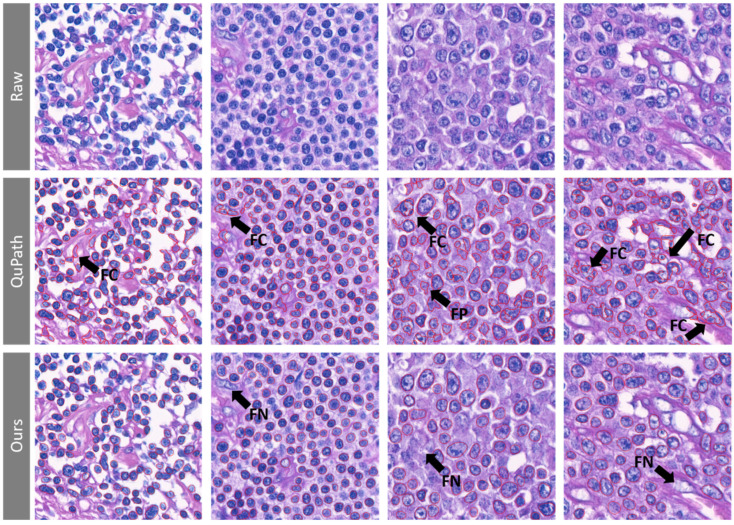
Comparison of contour segmentation results with the QuPath cell detection module. The first row represents the original image. For the QuPath cell detection module (the second row), default parameters were employed, except the nucleus sigma (radius of the nuclei) was set to 3 μm due to the pixel-to-μm ratio. Fragile contours (FC) and false positives (FP) occurred frequently in QuPath. Our lymphocyte detection model (the third row) showed fewer FC and FP but more false negatives (FN) at a score threshold of 0.2, along with an IoU threshold of 0.5, which might be due to the low confidence of the segmented contours.

**Table 1 cancers-13-05463-t001:** Clinicopathological features of the four borderline cases.

Case no./Sex/Age	T05A/Male/49	T31/Female/49	T60/Male/64	T63/Male/56
Tumor site	Ileum	Ileum	Jejunum and ileum	Ileum
Perforation	Present	Present	Present	Present
Cell size	Medium to large	Medium to large	Large	Medium to large
Morphology (mono vs. pleomorphic)	Pleomorphic	Pleomorphic	Mildly pleomorphic	Pleomorphic
CD8	+	+	+	+
CD30	-	-	-	-
CD56	+	+	+	+
TIA-1	-	+	+	+
Granzyme B	+	+	-	+
Lugano stage *	Stage IE	Stage IE	Stage IE	Stage IV
Follow-up (month)	DOD (34)	DOD (2)	DOD (6.5)	DOD (5.8)

Abbreviation: DOD, died of disease. * Stage IE: single extra-lymphatic site in the absence of nodal involvement.

**Table 2 cancers-13-05463-t002:** Dataset used at each phase of machine learning model.

	Lymphocyte Detection Model	Feature Extraction	T Cell Lymphoma Classification Model (XGBoost and CNN)
No. of cases (WSIs)	18	40	40
MEITL	14	26	26
ITCL-NOS	3	10	10
Borderline	1	4	4
No. of ROIs	33	400	400
ROI size	768 × 768 pixels	2304 × 2304 pixels (HPF)	2304 × 2304 pixels (HPF)
Data level	ROI-level	HPF-level	HPF-level
Data set	33 ROIs	400 HPFs (28 features per HPF)	36 cases (360 HPFs) 4 borderline cases were excluded
Training set	27 ROIs	-	3-fold cross-validation
Validation set	3 ROIs		
Testing set	3 ROIs		

Abbreviations: HPF: high-power field; ROI: region of interest.

**Table 3 cancers-13-05463-t003:** Definition of attributes in cellular morphometrics.

Attributes	Definition
Ratio of axis length	The ratio of the longest axis and the second-longest axis
Circularity	Ratio of overlapping pixels between the concentric circle and size of the cell
Entropy	Measure of the randomness of pixels in the cell
Area	Total numbers of pixels within the boundary of the cell.
Irregularity	Variance of length from the center of cell to each vertex of the boundary
Perimeter	Estimated total numbers of pixels along the cell boundary
Orientation	Cell orientation of the longest axis

**Table 4 cancers-13-05463-t004:** Comparison of model performance based on the case-level prediction.

Model	Model Comparison	XGB-1 (Morphology-only)	XGB-2 (Morphology + IHC)	CNN
AUC		0.966	0.955	0.820
95% CI		0.949–0.984	0.935–0.975	0.734–0.906
*p*-value (Delong’s Test)	XGB-1 VS. XGB-2	0.166		
	XGB-1 VS. CNN	0.003 *		
	XGB-2 VS. CNN	0.001 *		

Note: Model XGB-1: Only morphological features (28 feature scores per subject) were used as inputs; Model XGB-2: Both morphological features and immunohistochemical (IHC) phenotypes. * *p* < 0.05.

**Table 5 cancers-13-05463-t005:** Characteristics of the computed cellular morphometrics between cases of MEITL and ITCL-NOS. (* *p* < 0.05).

				Disease Type		*F*-Test	
			Overall	MEITL	ITCL-NOS	F-Statistics	*p*-Value
No.	Attributes	Moments (Mean ± SD)	*N* = 36	*N* = 26	*N* = 10		
1	Ratio of axis length	Mean	1.29 ± 0.06	1.27 ± 0.03	1.34 ± 0.06	11.53	<0.001 *
		Variance	0.04 ± 0.02	0.04 ± 0.01	0.06 ± 0.02	13.51	<0.001 *
		Skewness	1.52 ± 0.35	1.59 ± 0.22	1.48 ± 0.34	1.01	0.374
		Kurtosis	6.79 ± 2.80	7.18 ± 1.51	6.78 ± 2.50	0.6	0.557
2	Circularity	Mean	0.75 ± 0.03	0.76 ± 0.02	0.72 ± 0.03	10.25	<0.001 *
		Variance	0.01 ± 0.00	0.008 ± 0.001	0.010 ± 0.002	14.95	<0.001 *
		Skewness	−0.75 ± 0.26	−0.83 ± 0.19	−0.61 ± 0.27	3.64	0.037*
		Kurtosis	3.57 ± 0.70	3.68 ± 0.49	3.21 ± 0.69	2.86	0.071
3	Entropy	Mean	5.85 ± 0.10	5.86 ± 0.09	5.84 ± 0.10	0.31	0.734
		Variance	0.07 ± 0.01	0.07 ± 0.01	0.08 ± 0.01	4.24	0.023
		Skewness	−0.19 ± 0.14	−0.20 ± 0.09	−0.21 ± 0.10	2.84	0.072
		Kurtosis	3.39 ± 0.38	3.40 ± 0.22	3.38 ± 0.17	0.02	0.976
4	Nuclear area	Mean	41.72 ± 9.00	38.13 ± 7.25	45.84 ± 6.93	11.63	<0.001 *
		Variance	234.16 ± 155.71	157.74 ± 70.25	362.70 ± 118.91	24.09	<0.001 *
		Skewness	0.02 ± 0.01	0.03 ± 0.01	0.02 ± 0.01	2.27	0.119
		Kurtosis	0.14 ± 0.07	0.16 ± 0.07	0.12 ± 0.02	4.32	0.021 *
5	Irregularity	Mean	1.15 ± 0.40	0.94 ± 0.22	1.55 ± 0.30	22.18	<0.001 *
		Variance	1.75 ± 1.06	1.22 ± 0.40	2.87 ± 0.94	23.87	<0.001 *
		Skewness	0.45 ± 0.12	0.50 ± 0.08	0.40 ± 0.07	8.67	0.001 *
		Kurtosis	2.63 ± 1.57	3.16 ± 0.97	2.32 ± 1.00	7.14	0.003 *
6	Perimeter	Mean	24.32 ± 2.66	23.26 ± 2.23	25.64 ± 1.90	11	<0.001 *
		Variance	20.32 ± 9.91	15.33 ± 4.80	30.40 ± 6.42	25.97	<0.001 *
		Skewness	0.09 ± 0.07	0.11 ± 0.06	0.08 ± 0.03	3.35	0.047 *
		Kurtosis	0.66 ± 0.24	0.74 ± 0.19	0.55 ± 0.07	7.03	0.003 *
7	Orientation	Mean	1.55 ± 0.13	1.54 ± 0.12	1.57 ± 0.14	0.52	0.599
		Variance	0.71 ± 0.15	0.72 ± 0.14	0.69 ± 0.16	0.14	0.870
		Skewness	0.001 ± 0.004	0.0009 ± 0.07	−0.0002 ± 0.04	0.50	0.610
		Kurtosis	0.04 ± 0.01	0.04 ± 0.01	0.04 ± 0.01	0.62	0.543

## Data Availability

The datasets used and/or analyzed during the current study are available from the corresponding author on reasonable request.

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
