# Peer review of "Machine Learning Based on Morphological Features Enables Classification of Primary Intestinal T-Cell Lymphomas"

_cancers, 2021, doi:10.3390/cancers13215463_

Round 1
Reviewer 1 Report
In this manuscript, the authors present a machine learning approach to classify two subtypes of primary intestinal T-cell lymphomas such as MEITL and PTCL-NOS. Primarily, they segmented nuclei of lymphocytes using deep neural networks [(hybrid task cascade regional proposal convolutional network (HTC-CNN)] and calculated 28 features for each segmented nucleus of lymphocyte for quantitative analysis. Further, the computed nuclear morphometrics were used to make classification of the two subtypes using tree-based XGBoost algorithm. The proposed method achieves high accuracy and the work is interesting and useful in the field. However, they show the superiority of the proposed method by comparing with poor performance from CNN. The following concerns must be addressed before considering for publication.
- Data requirements for XGBoost and CNN model (such as Resnet50) to achieve optimum training are different. CNN is known to perform poorly where there is not sufficient data for training. Therefore, CNN model with some sort of data augmentation should have been compared to XGBoost model for a fair comparison.
- Reproducibility is an issue when any neural network model runs on a GPU. Any reproducibility criteria used for the CNN model during classification?
- Resnet50 and HTC-CNN model architecture is already built but hyperparameters such as Optimizer, activation function can be tuned for the dataset in hand. On the other hand, random search hyperparameter optimization was performed on the XGBoost model to know the best hyperparameters for the particular dataset.
- Any specific reason to use SGD optimizer with learning rate 0.001, batch size 16 for HTC-CNN?
- Any specific reason to use AdamW optimizer with learning rate 0.001, batch size 16 for Resnet50 CNN?
Reviewer 2 Report
The manuscript confuses the WHO classification of primary intestinal T cell lymphomas and the term PTCL-NOS which has been repeatedly used in this manuscript to describe a subtype of primary intestinal T cell lymphoma which is not recommended and unacceptable. According to the WHO classification and I will quote the WHO book as is on page 403: PTCL-NOS is a heterogeneous aggressive entity of systemic T cell lymphomas, while primarily most commonly affecting lymph nodes, may involve extranodal sites like skin and gastro-intestinal tract and In this setting, the diagnosis of PTCL, NOS, should be made only after more specific entities have been excluded""". That being said, the WHO recognizes intestinal T cell lymphoma, NOS Page:378"""This category is used for T-cell lymphomas arising in the intestines, or sometimes other sites in the gastrointestinal tract, that do not conform to either classic enteropathy associated T-cell lymphoma or monomorphic epitheliotropic intestinal T-cell lymphoma. Sometimes this diagnosis is made based on an inadequate biopsy in which the mucosal surface cannot be evaluated or immunophenotypic data are incomplete. It is not considered a specific disease entity.""REF: WHO. Page 378. ""At a recent workshop of the European Association for Hematopathology, most cases assigned to this category involved the colon. The cases were heterogeneous in their morphology and immunophenotype; 4 of the 5 cases with evaluable data were T-cell receptor–silent, but most had a cytotoxic phenotype. Several of the cases had widespread disease, so the intestines may not have been the primary site. All cases were clinically aggressive."""REF. WHO. Page 379.
In addition, if there is involvement of the gastrointestinal by a specific recognized entity by the WHO, then the lymphoma should be called as such examples "anaplastic large cell lymphoma", or "extranodal / extranasal NK/T cell lymphoma" etc...
IN my opinion, this manuscript is great in its principle idea and has potential benefits, however it need to be written without confusing the WHO classification and the authors need to clarify the 40 cases included in this study, where these small core biopsies insufficient for final diagnosis or excisional biopsies with sufficient tissue for final diagnosis? Why the authors excluded enteropathy associated T cell lymphomas and other T cell lymphomas involving the Intestine ? What do the authors mean by Borderline cases were collected from different hospitals, there is no such diagnosis neither in practice or in the WHO? Please: clarify Reference 6 was cited without explanation to its relevance to this study while reference 28 was cited with relevance to morphology growth pattern which is acceptable.
Author Response
Reply:
We thank the reviewer for these very important comments in using the current WHO terms for specific diagnoses. In the current WHO scheme, four entities are listed for primary intestinal T-cell lymphomas. They are, as you are aware, EATL, MEITL, PTCL-NOS, and indolent T-cell LPD of the GI tract. In our study cases, all were surgical specimens with staging workup confirmed primary intestinal T-cell lymphoma. There were literally no EATL in Taiwan as shown in our previous study (detailed in ournext reply). Most of the cases were MEITL; however, there were some cases with pleomorphic, medium to large-sized nuclei and an immunophenotype atypical for the current criteria of MEITL (medium-sized cells with round and regular nuclei with expression of both CD8 and CD56). In the current WHO scheme, these cases have to be placed under the diagnosis of primary intestinal PTCL-NOS; but as clarified in the WHO book, this is not considered as a specific entity.
One of use (SSC) happens to be a coauthor in the section of MEITL in the forthcoming 5th edition of WHO Hematolymphoid book. In the manuscript that is currently under preparation, the tumor types of primary intestinal T-cell lymphoma remain the same. SSC is the responsible (first) author for the ENKTL chapter extranodal NK/T-cell lymphoma (ENKTL) in the 4th WHO edition of Head and Heck and for the next (5th) WHO edition of Hematolymphoid tumors. In the manuscript that is currently ready, we coauthors of this ENKTL chapter recognized that the majority (80%) of ENKTL occurs in the nasal region, and these cases are considered nasal type (a clinical but not pathological subtype). The remaining 20% cases occur in the skin, GI tract, testis and other rare sites. These latter cases are considered non-nasal or extra-nasal clinical subtype. The ENKTL cases mentioned in our manuscript were primary intestinal ENKTL, i.e., the non-nasal or extra-nasal subtype of ENKTL.
I am sure the reviewer is very familiar with both ALK+ and ALK- anaplastic large cell lymphoma (ALCL). Primary ALCL may occur in the lymph nodes and extranodal sites. Primary gastrointestinal ALCL is rare. One of us (SSC) have reported a series of primary gastrointestinal ALCL (Pathology. 2017 Aug;49(5):479-485; PMID 28693479). In that series of nice cases (seven from the collection of SSC and two from Japan), two cases were positive for ALK and the remaining cases were negative.
In a recent publication in Cancers, Kim et al reported a retrospective study of primary intestinal T and NK cell lymphoma from Korea (Cancers (Basel) 2021 May 29;13(11):2679), the most common histologic type was extranodal NK/T-cell lymphoma, nasal type (ENKTL; 34%), followed by monomorphic epitheliotropic intestinal T-cell lymphoma (MEITL; 32%), intestinal T-cell lymphoma, NOS (ITCL, NOS, 18%), anaplastic large cell lymphoma, ALK-negative (ALCL, ALK-; 13%). This study highlights the presence of primary intestinal ENKTL, PTCL-NOS (or ITCL-NOS), and ALCL as mentioned above.
I hope we can convince the reviewer that we were stringent in the classification of our primary intestinal T-cell lymphoma cases. We have revised our manuscript to make sure that we clearly categorized our cases according to the current WHO blue book. In order to stress that the PTCL-NOS in our study were primary intestinal cases, we have replaced “PTCL-NOS” in the revised manuscript by intestinal T-cell lymphoma, NOS (ITCL-NOS).”
The followings are our point-to-point reply to the specific comments:
- All the 40 specimens are surgical specimens, not small core biopsies.
- There are essentially no patients with celiac disease in Taiwan; and hence, no celiac disease-associated enteropathy associated T cell lymphomas (EATL) cases. In our previous study of 30 cases of primary intestinal T or NK cell lymphoma, all seven cases examined for serum anti-tissue transglutaminase by ELISA were all negative, arguing against EATL (Am J Surg Pathol 2009;33:1230–1240). Among the 40 cases in the current study, four of the seven cases from the previous study were included. They were Cases no. T02, T13, T18, and T20. They were negative for serum anti-tissue transglutaminase.
- Systemic survey for staging confirmed that these cases were all primary intestinal T-cell lymphoma, not secondary involvement by lymphomas from other primary site.
- We collected primary intestinal T-cell lymphoma from other hospitals in Taiwan and then tried to classify the cases according to the WHO classification scheme. As the reviewer is very aware that the major entities of primary intestinal T-cell lymphoma are EATL and monomorphic epitheliotropic T-cell lymphoma (MEITL), while those with atypical features not fitting into either of these two entities are put into the wastebasket of primary intestinal peripheral T-cell lymphoma, not otherwise specified (PTCL-NOS). Most of the outside cases were diagnosed either as MEITL or PTCL-NOS (more specifically, intestinal T-cell lymphoma, not otherwise specified or ITCL-NOS). We examined the histological features of the tumor cells (monomorphic vs. pleomorphic) and correlated the immunohistochemical findings (expression of CD8, CD56, cytotoxic markers, and in situ hybridization for EBV-encoded mRNA (EBER)). We found that the tumor cells of four cases in this study (Table 1) were larger and more pleomorphic as compared to the WHO criteria of medium-sized cells with round nuclei. However, the immunophenotype was typical for MEITL. They were all negative for EBER. That’s the reason we called these four cases borderline.
- Reference no. 6 (Am J Surg Pathol 2009;33:1230–1240) is the prior study that we just mentioned. We cited that paper to stress that in contrast to the medium-sized, round nuclei in MEITL, the neoplastic cells of primary intestinal PTCL-NOS, i.e., ITCL-NOS, were usually medium to large-sized and pleomorphic.
Reviewer 3 Report
Machine learning based on morphological features is feasible in classifying primary intestinal T-cell lymphomas
The authors presented a machine learning approach for accurate quantification of nuclear morphometrics and differential diagnosis of primary intestinal T-cell lymphomas. They tried to inspect the feasibility of ML techniques based on morphological features in classifying two subtypes of primary intestinal T-cell lymphomas (PITLs): monomorphic epitheliotropic intestinal T-cell lymphoma (MEITL) versus peripheral T-cell lymphoma, not otherwise specified (PTCL-NOS).
They used 40 slides to train extreme gradient boosting (XGBoost) and with that they performed classification.
They used model to accomplish this, and their results are interesting
The article is well-written and interesting. I personally like this study.
Authors managed to modify their article considerably.
Minor comments:
- The contribution of this paper with the respect to dimensionality reduction needs to be highlighted. It is important that show how to deal with high imaging information while you had 40 cases, some examples: doi: 10.1097/CCM.0000000000000878, doi: 10.1038/s41598-021-88239-y, doi: 10.1093/annonc/mdx034.
- Moreover, there is a need for clarification regarding the morphological throughput extracted from regions of interest.
- Please clarify why F-test is used to evaluate the statistical significance of features.
- Please increase the quality of your figures. For example, figures 1,2 need to be modified.
Round 2
Reviewer 2 Report
Abstract (fourth sentence) and in the introduction : I recommend remove the word peripheral and keep it as intestinal T cell lymphoma, NOS (ITCL, NOS) like the wording of the WHO.
Table page 3 : T63; Tumor site= correct Ilum to Ileum
Author Response
Response to reviewer comments:
Thanks for the kindly remind from the reviewer. We have removed the word "peripheral" in the abstract (line 4) and the same condition happened in the introduction (line4). Also, the spelling error in Table1 has been corrected to "Ileum".
The English editing has also been processed for better readability. Again, we deeply appreciate your precious comments and suggestions.